# Molecular Characterization of Subdomain Specification of Cochlear Duct Based on Foxg1 and Gata3

**DOI:** 10.3390/ijms252312700

**Published:** 2024-11-26

**Authors:** Yongjin Gil, Jiho Ryu, Hayoung Yang, Yechan Ma, Ki-Hoan Nam, Sung-Wuk Jang, Sungbo Shim

**Affiliations:** 1Department of Biochemistry, Chungbuk National University, Cheongju 28644, Republic of Korea; kilyj@chungbuk.ac.kr (Y.G.); yhs9736@naver.com (J.R.); hayoung.yang10@gmail.com (H.Y.); 2Department of Biochemistry and Molecular Biology, Brain Korea 21 Project, Asan Medical Center, University of Ulsan College of Medicine, Seoul 138-736, Republic of Korea; dpcks9867@gmail.com; 3Laboratory Animal Resource and Research Center, Korea Research Institute of Bioscience and Biotechnology, Cheongju 28116, Republic of Korea; namk@kribb.re.kr

**Keywords:** inner ear, cochlear duct, Foxg1

## Abstract

The inner ear is one of the sensory organs of vertebrates and is largely composed of the vestibule, which controls balance, and the cochlea, which is responsible for hearing. In particular, a problem in cochlear development can lead to hearing loss. Although numerous studies have been conducted on genes involved in the development of the cochlea, many areas still need to be discovered regarding factors that control the patterning of the early cochlear duct. Herein, based on the dynamic expression pattern of FOXG1 in the apical and basal regions of the E13.5 cochlear duct, we identified detailed expression regions through an open-source analysis of single-cell RNA analysis data and demonstrated a clinical correlation with hearing loss. The distinct expression patterns of FOXG1 and GATA3 during the patterning process of the cochlear duct provide important clues to understanding how the fates of the apical and basal regions are divided. These results are expected to be extremely important not only for understanding the molecular mechanisms involved in the early development of the cochlear duct, but also for identifying potential genes that cause hearing loss.

## 1. Introduction

Hearing loss is one of the most common sensory deficits in humans. It can occur at various ages, and its severity is classified according to the degree of symptoms. Sensorineural hearing loss (SNHL), which is caused by damage to the auditory nerve (cranial nerve VIII) and the central nervous system, is the most common type of hearing loss and is usually permanent [1,2,3]. SNHL is caused by environmental factors (e.g., aging, infectious diseases, noise exposure, and drug abuse) and genetic factors (e.g., genetic mutations); the majority of cases of congenital hearing loss are caused by genetic mutations [4]. Approximately 200 deafness-related loci and causative genes have been reported [5], and a pathological analysis is being conducted using animal models that can reproduce the human phenotype [6].

The inner ear is one of the representative sensory organs of vertebrates and is composed of the vestibule, which controls balance, and the cochlea, which recognizes sound [7]. The vertebrate inner ear is differentiated from the otic vesicle that occurs through the invagination of the ectoderm near the hindbrain. The otic vesicle is influenced by various factors from adjacent tissues to have an identity in each region along the three-dimensional axis [8]. The representative factors that determine this specification include WNT; sonic hedgehog (SHH), which determines the dorsal–ventral axis; retinoic acid, which forms the anterior–posterior axis; and *Fgf3*, which affects the medial–lateral axis [9,10,11]. Afterward, it undergoes sophisticated morphogenesis to differentiate into the inner ear, with the dorsal region extending to the vestibule and the ventral region developing into the cochlea [12]. The epithelium of the developing cochlear duct can be largely divided into the roof and the floor and is categorized into the following five detailed regions in the early stage of development: Reissner’s membrane and stria vascularis that differentiate from the roof; the prosensory domain (PSD) that differentiates from the floor and becomes the organ of corti (OC); a cellularly dense medial domain, the greater epithelial ridge (GER); and a less dense lateral domain, the lesser epithelial ridge (LER). This region specification is already completed during the early stage of cochlea duct outgrowth [13].

Numerous studies have reported on factors important for the formation of these regions. For example, *Otx2* is expressed in the ventral portion of the otic vesicle and then in the roof of the cochlea duct [14]. *Otx2* loss in the cochlear duct leads to the loss of the roof fate and the induction of the ectopic PSD [15]. *Jag1* is expressed in the GER, including the PSD, early in development [16] and has an extremely important function in the differentiation of sensory progenitor cells into hair and supporting cells [17,18,19]. *Sox2* is also expressed in the PSD and is essential for sensory cell differentiation [20,21,22]. *Gata3*, known as a mutant gene for HDR syndrome, is expressed in the floor part of the cochlear duct, particularly in the lateral region [23], and *Gata3* loss during development results in the loss of neurosensory cells and the failed formation of normal sensory organs [24,25]. *Bmp4* is expressed in the LER [26,27], and a decrease in it in the LER results in the underdevelopment of the PSD and the ectopic expression of GER marker genes [28]. *Bmp4*+/− mice have been reported to have partial hearing loss due to structural problems in the inner ear and a decrease in neurites in the OC [29].

*Foxg1*, a member of the forkhead family, is known to be an extremely important transcription factor in nervous system development and is expressed in the early otic vesicle [30]. The early expression of *Foxg1* in the otic vesicle can be regulated by the SoxE family [31]; *Foxg1* is also regulated by several inner-ear development factors, such as *Six1*, *Gata3*, and *Sox2* [32]. During the inner-ear development of *Foxg1* knockout mice, vestibular defects, shortened cochlear ducts, and an abnormal increase in the number of hair cells and impaired polarities were observed [33,34]. A functional study of *Foxg1* in relation to hair cell differentiation at the postnatal stage reported that *Foxg1* is important for hair cell maintenance as it regulates apoptosis [35]; in hair cell-specific *Foxg1* cKO mice, the number of hair cells was greatly reduced and hearing was impaired due to increased apoptosis [36]. In supporting cell-specific *Foxg1* cKO mice, the phenomenon of supporting cells trans-differentiating into hair cells has been reported, which appeared to be deeply related to the possibility of hair cell regeneration treatment [37]. Compared with such studies on normal differentiation and hair cell maintenance, studies on the detailed expression patterns and functions of *Foxg1* in the early developmental stages are lacking. This is because these studies are difficult to conduct due to the severe morphological defects and the high lethality of the inner ear of *Foxg1* KO mice [30,36]. In the present study, we discovered a unique expression pattern of FOXG1 in E13.5, the early developmental stage of the cochlear duct. We aimed to organize this in detail and identify the detailed cell types in which *Foxg1* is actually expressed using open-source single-cell analysis data.

## 2. Results

### 2.1. Dynamics of FOXG1 Expression Patterns in the Developing Cochlear Duct

To investigate the expression pattern of Foxg1 in the developing cochlear duct, immunohistochemistry was employed so as to observe the detailed expression pattern according to the developmental stage. Gata3 is a well-known gene that is extremely important in inner-ear development and is expressed in the otic vesicle, hair cells of the OC, supporting cells, and surrounding spiral ganglion neurons (SGNs) [25]; thus, it was co-stained with FOXG1 (Figure 1). FOXG1 and GATA3 were strongly expressed throughout the otic vesicle at E9.5 (Figure 1A). Furthermore, they were expressed throughout the otic vesicle at E10.5, but FOXG1 was expressed more strongly in the ventral part and GATA3 in the mid-ventral part (Figure 1B). The cochlear duct began to extend from the ventral region of the otic vesicle at E11.5 [38]. At E11.5, GATA3 expressed bias toward the medial–dorsal part, whereas FOXG1 was expressed throughout (Figure 1C). At E12.5, GATA3 was expressed in the floor region of the dorsal part of the cochlear duct, whereas FOXG1 was still expressed in the roof and floor (Figure 1D). At E13.5, the cochlear duct was extended to about one turn, and one duct section near the apex and two duct sections toward the base could be seen in the section. GATA3 exhibited a consistent pattern of expression in the floor part, whereas FOXG1 was differently expressed depending on the location, i.e., it was expressed throughout the apical duct and exclusively together with GATA3 in the basal duct (Figure 1E). In particular, in the floor region of the cochlear duct, the expression area of FOXG1 was restricted and reduced at the base compared to the apex (Figure 1F). As the FOXG1 expression began to be restricted to more detailed regions in the cochlear duct at E13.5, the FOXG1 expression in the serial sections of the E13.5 cochlear duct was confirmed to determine the precise expression pattern (Appendix A). Both FOXG1 and GATA3 were expressed in SGNs, and in most sections of the cochlear duct epithelium, FOXG1 and GATA3 exhibited an exclusive expression pattern. However, in the apex duct, FOXG1 was expressed throughout the duct, including the GATA3 expression area (Appendix A). This indicates that the cochlear duct at E13.5 has different developmental states from base to apex, which is similar to the pattern of hair cell differentiation. The cochlear duct is already a complex structure with detailed regions for differentiation, and FOXG1 is expressed throughout the apical duct even at E13.5; thus, it is thought that it does not affect region determination. However, the FOXG1 expression appears to be sensitively regulated according to a specific developmental period. Therefore, it is important to confirm in which detailed region of the cochlear duct FOXG1 is expressed.

### 2.2. Identification of Inner-Ear Cell Types Using Public Data

Numerous cell subtypes have been identified in previous single-cell RNA sequencing on the inner ear [39]. Among them, we isolated cells related to the cochlear duct and aimed to identify genes related to patterning involved in normal cochlear duct development, including *Foxg1* and *Gata3*. To this end, single-cell RNA sequencing data defined by the auditory epithelial trajectory of E9.5–E13.5 mice and single-cell RNA sequencing data extracted from the inner ear of *Foxg1Cre*-Ai9 embryos were integrated using the harmony method (Figure 2B,C) [39,40,41]. A total of 81,102 inner-ear cells underwent a rigorous quality control process (Appendix A). Eight cell subtypes were defined via unsupervised clustering and marker gene expression and visualized via UMAP projection (Figure 2A,D,E). Specifically, they were clustered into osteoblast subtypes (C0) expressing *Col1a2*, *Col2a1*, and *Fstl1* [42]; radial glial progenitor cell subtypes (C1) expressing *Sox2* and *Hes5* [43]; cochlear epithelial cell subtypes (C2) expressing *Epcam* [44]; neuronal cell subtypes (C3) expressing *Tubb3* and *Dcx* [45,46]; inner-ear glial cell subtypes (C4) expressing *Plp1* [47]; blood cell subtypes (C5) expressing Hba- and Hbb-families [48]; endothelial–hematopoietic cell subtypes (C6) expressing *Kdr*, *Cdh5*, and *Ramp2* [49]; and macrophage subtypes (C7) expressing *Fcer1g* [50]. This clustering closely reflected the cell types previously reported to exist within the inner ear.

### 2.3. Subclustering of Cochlear Epithelial Cell Subtypes

To elucidate the exclusive expression patterns of FOXG1 and GATA3, E13.5 cochlear epithelial cell subtypes (C2) were subsetted, and then harmony integration was performed (Appendix A) [41]. A total of 2095 cells were subsetted, and subclustering was performed at various resolutions to distinguish specific areas exhibiting patterning. At a resolution of 0.03, 2 clusters were identified: at 0.1, 6 clusters; and at 0.5, 9 clusters (Appendix A). When the resolution was set to 0.5, marker genes, such as Tecta (C0), a cochlear medial side marker; Oc90 (C1), an otic marker; Clu, Fst, and Gata3 (C2), vestibular lateral side markers; and Hey2, Sox2 (C3), and Bmp4 (C8), LER markers, were identified (Appendix A) [51]. However, these marker genes overlapped in multiple clusters and actually exhibited a broader expression pattern (Appendix A). This is probably because differentiating cells exhibit transcriptome dynamics [52].

### 2.4. hdWGCNA of Cochlear Epithelial Cell Subtypes

Because *Foxg1* and *Gata3* showed extremely broad expression patterns on UMAP, it was difficult to clearly distinguish the exact *Foxg1* and *Gata3* expression areas. To address this problem, we employed a method to classify genes with similar expression patterns that are thought to be involved in specific biological functions. For this, high-dimensional weighted gene co-expression network analysis (hdWGCNA) was conducted on E13.5 cochlear epithelial cell subtypes [53]. To set the soft-thresholding power to secure the scale-free topology of the network, the scale-free topology model fit was evaluated for various power values. The optimal soft power threshold for the model fit value exceeding 0.8 was 14 (Appendix A). Based on this, network analysis was conducted, and 10 modules were identified from the dendrogram (Appendix A, Figure 3A). By confirming the eigengene-based connectivity (kME) of each gene, we found hub genes representing the modules and divided these modules according to specific areas. As many marker genes whose expression domains are well known were included as hub genes in the modules, the modules could be classified according to the region of the cochlear duct. Specifically, CD−M1 containing *Oc90*, *Otx2*, and *Otx1*, which are roof/epithelial markers; CD−M2 containing *Clu*, *Fst*, and *Gata3*, which are LER markers; CD−M3 containing *Sox2* and *Hey2*, which are PSD markers; *Tecta*, *Tectb*, *Fgf10*, and *Jag1*, which are GER markers; CD−M9 containing ribosome-related genes; and CD−M10 containing mitochondria-related genes were classified (Figure 3A, Appendix A) [51].

The marker genes representing these modules exhibited high connectivity within the modules (Figure 3B). Gene Ontology (GO) enrichment analysis was conducted to confirm and classify the biological functions of the remaining modules. For CD−M4, ontology terms related to cell cycle and genes such as *Ube2c*, *Top2a*, and *Mki67* were included. For CD−M5, ontology terms such as “pattern specification process”, “regionalization”, and “epithelial tube morphogenesis” as well as genes such as *Irx1*, *Pax2*, *Tbx3*, and *Lrp2* were included, and the expression was shown in roof. For CD−M6, ontology terms related to “epithelial tube morphogenesis” as well as cytoskeleton and genes such as *Rgcc*, *Gsn*, *Epha7*, *Six2*, and *Foxg1* were included, and the expression was shown in GER and roof. For CD−M7, ontology terms related to *Catilage*, *Prrx1*, and multiple collagen genes were included, whereas for CD−M8, ontology terms related to protein and RNA metabolism as well as genes such as *Hspa8*, *Npm1*, *Rbmxl1*, and *Rbm8a* were included. Analysis based on these modules enabled the classification of gene groups according to specific biological functions as well as the patterning of the developing cochlear duct.

### 2.5. Network Analysis of FOXG1 and GATA3

We attempted to confirm the expression regions of *Foxg1* and *Gata3* using modules defined according to distinct regions and functions. Similarly to the IHC results, on UMAP, *Foxg1* exhibited an exclusive expression pattern in roof and GER corresponding to CD−M6 and CD−M3, whereas *Gata3* showed high connectivity in the corresponding modules (Figure 3B and Figure 4A). Next, by analyzing genes co-expressed with *Foxg1* and *Gata3*, we conducted correlation analysis to identify the genes involved in similar biological processes and constructed a network. Genes corresponding to the *Foxg1* network were included in CD−M6 and CD−M3, whereas those corresponding to the *Gata3* network were included in CD−M2 (Figure 4B). GO analysis was conducted to confirm the biological processes associated with the genes included in each network (Figure 4D,E). In the *Foxg1* network, which includes genes such as *Tecta*, *Fgf10*, *Bcl2*, and *Six2*, ontology terms related to ear development and epithelial formation were identified [54,55,56,57]. In the *Gata3* network, which includes genes such as *Isl1*, *Sfrp1*, *Nr2f2*, and *Efnb2*, ontology terms related to epithelial cell proliferation and migration were identified [58,59,60,61], which were similar to the GO results of CD−M2 (Appendix A).

### 2.6. Expression of Various Marker Genes in the E13.5 Cochlear Duct

The detailed expression area of *FOXG1* in the developing cochlear duct was identified through module analysis. The reliability of the analysis was examined by confirming the expression of well-known marker genes among the hub genes in other modules in the E13.5 cochlear duct. The cochlear duct roof marker OTX2, which is included in Module 1, shows that the expression location of FOXG1 is gradually narrowed to the GER, including part of the roof (Figure 5A). GATA3, which is exclusively located there, is expressed in the LER of the floor part, and the PSD located between the LER and the GER was confirmed through the expression of the corresponding marker, namely SOX2 (Figure 5B) [62]. JAG1, which corresponds to Module 3, containing the GER marker genes, was also expressed in the GER where PSD and FOXG1 are expressed, as previously described (Figure 5D) [63]. Although Sox9 was not classified as a module-delimiting gene, it was confirmed to have been expressed throughout the cochlear duct epithelium, as in a previous study (Figure 5C and Appendix A) [64].

### 2.7. The Relationship Between Hearing Loss-Associated Disease and Modules

To confirm the clinical relevance of actual hearing loss with modules classified according to the patterning and specific biological functions of the developing inner ear, we classified genes related to hearing loss in ClinGen according to the genes included in each module. In roof/epithelial (CD−M1), *Myo6* and *Actg1* were associated with nonsyndromic hearing loss (NSHL), *Hsd17b4* with Perrault syndrome, *Ednrb* and *Edn3* with Waardenburg syndrome, and *Actg1* with Baraitser–Winter syndrome. In LER (CD−M2), *Ush1c*, *Otogl*, *Gjb2*, and *Eps8* were associated with NSHL; *Ush1c* and *Clrn1* with Usher syndrome, Six1 with branchiootorenal syndrome; *Gjb2* with skin phenotype and HL; *Gata3* with hypoparathyroidism, sensorineural hearing loss, and renal dysplasia; and *Col9a2* with Stickler syndrome. In GER (CD−M3), *Tecta*, *Pcdh15*, and *Cdc14a* were associated with NSHL, *Slitrk6* with deafness and myopia, *Pcdh15* with Usher syndrome, *Eya1* with branchiootorenal syndrome, *Chd7* with CHARGE syndrome, and *Cdc14a* with hearing impairment and infertile male syndrome. In cell cycle (CD−M4), *Tubb4b* was associated with Leber congenital amaurosis, *Dnmt1* with DNMT1 methylopathy, and *Diaph3* with auditory neuropathy spectrum disorder. In epithelial patterning (CD−M5), *Tmtc2*, *Hgf*, and *Ccdc50* were associated with NSHL and *Ror1* with hearing loss and auditory neuropathy. In GER and roof (CD−M6), *Otoa*, *Gsdme*, and *Espn* were associated with NSHL. In the catalog (CD−M7), *Pou3f4*, *Homer2*, and *Col11a2* were associated with NSHL, *Snai2* with Waardenburg syndrome, *Lars2* with Perrault syndrome, and *Col11a2* with otospondylomegaepiphyseal dysplasia. In the ribosome (CD−M9), *Crym* was associated with NSHL. Protein and RNA metabolism (CD−M8) and mitochondria (CD−M10) did not contain any genes related to hearing loss (Figure 6A,C, Appendix A). The genes included in the modules related to hearing loss exhibited high connectivity in the modules (Figure 6B).

## 3. Discussion

The vertebrate inner ear is formed into a complex structure through the expression regulation of numerous factors from the early developmental stage to the completion of differentiation. *Foxg1* is expressed in the telencephalon and otic vesicle from the early developmental stage and plays a pivotal role in the development of the central nervous system [30]. Its expression is continuously maintained in the adult inner-ear cochlea [33]. *Foxg1*-Cre mice have been widely used in functional studies of genes in the inner ear owing to the otic epithelium-specific expression pattern in the early developmental stage [65]. However, detailed descriptions of the expression pattern of FOXG1 in the developing cochlear duct have not yet been reported. 

This study demonstrates that FOXG1 is expressed throughout the cochlear duct during early development and then becomes increasingly restricted to the GER, which becomes the inner sulcus, at E13.5. Furthermore, unlike the basal and middle ducts, it maintained overall expression in the apical duct and exhibited a dynamic pattern. Through this FOXG1 expression pattern, it can be predicted that the developmental speed within the cochlear duct is different and that differentiation sequentially occurs along the base–apex axis. The hair cell differentiation process is also affected by the ATOH1 and SHH gradients and occurs along the base–apex axis [3,66,67]. As the patterning along the medial–lateral axis of the cochlear duct is completed at an earlier stage than E13.5, FOXG1 is unlikely to function as a cell fate determinant. As FOXG1 has different functions in neural progenitor cells and postmitotic neurons in cerebral neurogenesis [68,69,70,71], it is thought to have different functions when expressed overall in the duct and when expressed restrictedly to the GER. In fact, as Foxg1 expression continues from the early otic placode, Foxg1 null mice demonstrate a wide range of morphological defects, such as vestibular dysfunction, shortened cochlea, and hair cells with abnormal number and impaired polarity [33,34]; however, the detailed mechanisms have been difficult to elucidate. Furthermore, hair cell-specific Foxg1-cKO mice exhibited a phenomenon opposite to that of null mice, in which hair cell apoptosis increased [36]. This indicates that Foxg1 performs different functions at different times. Thus, studies on how Foxg1 expression is regulated in a stage- and cell type-specific manner are warranted to elucidate its various functions. In particular, the mutually exclusive expression patterns of Gata3 and Foxg1 at E13.5 identified in this study may provide a new direction for studies on hair cell maturation and regeneration in addition to hair cell differentiation [72].

E13.5 cochlear epithelial cells were classified into gene groups with similar expression patterns via hdWGCNA (Figure 3A). This enabled us to identify a specific region expressing *Foxg1*. *Foxg1* was classified into CD−M6, and genes expressed in the GER and roof were included in the module. In the classified modules, we constructed a co-expression gene network centered on *Foxg1* and *Gata3*, respectively. Ebf1, one of the genes included in the Foxg1 network, is mainly expressed in the same GER region as Foxg1; although it has not been reported in humans, an increase in the number of hair cells and supporting cells in the cochlea of Ebf1 KO mice as well as hearing loss has been reported [73,74]. As similar defects, such as an abnormal increase in hair cells, have been reported in Foxg1 KO mice [33], genes classified into the same network are likely to share similar expressions and functions. Furthermore, Isl1, one of the genes included in the Gata3 network, has been reported to have the potential to act as a transcriptional upstream regulator of Gata3 as it downregulated Gata3 in the developing inner ear of cKO mice [58]. This network model makes it easier to predict the molecular architecture at the genome-wide level among genes with similar expression patterns. Among the hearing loss-associated genes reported in the patient, 35 genes included in the module were extracted and classified, but there was no specific syndrome or characteristic associated with the module. Moreover, the pairwise correlation analysis of the 35 hearing loss-associated genes was conducted (Appendix A), and many genes with high mutual correlation were included in CD−M2 and CD−M3. As the genes of CD−M2 and CD−M3 are expressed in LER, GER, and PSD, it is highly likely that they co-express and act as key factors in the direct hair cell differentiation process. Through this module analysis method, it is expected that the expression area of the hearing loss genes discovered later can be mapped during the developmental process, and it will be helpful in understanding the disease occurrence mechanism through the related gene network. 

## 4. Materials and Methods

### 4.1. Animals

All animal experiments were conducted in accordance with the protocols approved by the Institutional Animal Care and Use Committee of the Asan Institute for Life Sciences (No. 2018-12-256). ICR (CD1) mice were used in the analysis.

### 4.2. Immunohistochemistry and Fluorescent Quantification Analysis

Whole embryo (E9.5–E13.5) was fixed in 4% paraformaldehyde in 1 × phosphate-buffered saline (PBS) at 4 °C overnight. The samples were embedded in 2% agarose/PBS gel and sectioned at 70 μm thickness using a vibratome (VT1000S, Leica Biosystems, Germany). After washing with PBS, the sections were incubated with blocking buffer (5% normal donkey serum, Jackson ImmunoResearch Labs, West Grove, PA, USA), 0.3% TritionX-100 (Sigma-Aldrich, Burlington, MA, USA), 0.2% lysine (Sigma-Aldrich), 0.2% glycine (Sigma-Aldrich), and 1% bovine serum albumin for 60 min at RT. Then, the slices were incubated with a primary antibody recognizing FOXG1 (rabbit, 1:200, ab196868, Abcam, Cambridge, UK), GATA3 (goat, 1:150, AF2605, R&D systems, Minneapolis, MN, USA), Jagged-1 (mouse, 1:100, sc-390177, Santa Cruz Biotechnology, Dallas, TX, USA), SOX2 (rabbit, 1:200, AB5603, Merck Millipore, Burlington, MA, USA), SOX9 (rabbit, 1:1000, AB5535, Merck Millipore), and OTX2 (goat, 1:500, AF1979, BD biosciences, Franklin Lakes, NJ, USA) in blocking buffer and with the appropriate secondary antibody (Jackson ImmunoResearch Labs). Images were captured using a C2 confocal microscope (Nikon, Tokyo, Japan).

To quantitatively compare the changes in the expression areas of Foxg1 and Gata3 in E13.5 cochlear ducts, five sections containing both apical and basal ducts were utilized. Using ImageJ (v.1.53), the areas corresponding to the roof and floor of each section were designated as ROI, and the threshold for each channel was set to calculate the %Area value. The quantitative analysis of these values was conducted using Student’s *t*-test and rstatix (v.0.7.2), followed by Bonferroni correction; visualization was performed using ggpubr (v.0.6.0) [75,76].

### 4.3. Dataset of Inner-Ear Samples

The inner ears were extracted from previously reported *Foxg1Cre*-Ai9 embryos, and single-cell RNA sequencing was performed, including one E9.5 day, three E11.5 day, and four E13.5 day independent biological replicates. All raw filtered counts matrices were deposited in the Gene Expression Omnibus (GEO) with the accession number GEO: GSE178931 [39]. To confirm organogenesis, single-cell RNA sequencing was performed from 61 embryos between E9.5 and E13.5 days, containing approximately 2 million cells. Among the 56 subtrajectories, 5000 cells were classified as auditory epithelial trajectories through marker genes, such as *Epcam*, *Otol1*, *Oc90*, and Nox3. The CDS object of monocle3 that passed the filtering process was deposited in the Mouse Organogenesis Cell Atlas (https://oncoscape.v3.sttrcancer.org/atlas.gs.washington.edu.mouse.rna/downloads, accessed on 26 September 2024) [40]. The overall data analysis sequence is diagrammed in Appendix A.

### 4.4. Preprocessing, Clustering, and Cell Type Identification

Quality control and preprocessing of single-cell RNA sequencing data were performed using Seurat (v.5.1.0) [77]. Raw count matrix and CDS objects were converted to Seurat objects and then merged. Subsequently, the data quality was evaluated using the total RNA molecule count per cell, number of unique genes detected per cell, and percentage of RNA counts from mitochondrial genes. The total RNA molecule count per cell ranged from 366 to 141,411, with a median of 9078. The number of unique genes detected per cell ranged from 35 to 9885, with a median of 3036, and the percentage of RNA counts from mitochondrial genes ranged from 0 to 97.14, with a median of 4.63. Because low-quality cells were included, they were filtered according to the following criteria: (1) total RNA molecule count per cell less than 800, (2) number of unique genes detected per cell less than 200, and (3) percentage of RNA counts from mitochondrial genes greater than 10%. Consequently, 89,399 cells were filtered into 81,102 cells. The filtered UMI counts were normalized using Seurat’s SCTransform function as default parameters to remove the technical variability caused by sequencing and clearly separate cell types [78]. Then, the harmony method, which effectively preserves inter-cell heterogeneity for integration between replicates and batch correction, was employed [41]. Before integration, the RunPCA function was used to reduce the dimension by setting the number of principal components to 30. Integration was performed using the IntegrateLayers function after selecting 3000 integration features using the SelectIntegrationFeature function. Otic epithelial cells were identified by more clearly separating cell heterogeneity using the SCTransform and harmony methods [79]. After performing harmony integration, Seurat’s clustering functions were used. Specifically, the FindNeighbors function was used with dims = 1:30 as a parameter. Then, the FindClusters function was used with a resolution of 0.03 as a parameter to identify 8 clusters. Finally, the RunUMAP function was used with reduction = “harmony”, dims = 1:30 as a parameter to visualize the integrated data in a low-dimensional space. To identify the marker gene of each cluster in the SCT assay, the PrepSCTFindMarkers and FindAllMarkers functions were used. Each cluster was classified into cell types based on previously known marker genes.

### 4.5. Subclustering of Otic Epithelial Cell Subtypes

After subsetting the E13.5-day otic epithelial cell subtype (C2), the same preprocessing as previously described was performed. When the resolution of the FindClusters function was set to 0.03, 2 clusters were identified; at a resolution of 0.1, 6 clusters were identified; whereas at 0.5, 9 clusters were found. Marker genes were identified in the nine clusters at a resolution of 0.5, as in the previous method.

### 4.6. High-Dimensional WGNCA Analysis

Using hdWGCNA (v.0.3.03), co-expression network analysis was conducted on the E13.5-day otic epithelial cell subtype [53]. After setting the otic epithelial cell subtype using the SetupForWGCNA function, the SCT assay was set using the SetDatExpr function. Next, to infer an appropriate soft power threshold, the TestSoftPowers function was used to perform network type in a signed manner. After calculating the lowest soft power threshold for which the scale-free topology model fit is 0.8 or higher, the ConstructNetwork function was used to construct a co-expression network. Next, after scaling using Seurat’s ScaleData function, the harmonized module eigengenes were computed using the ModuleEigengenes function. To identify the hub gene of each module, the ModuleConnectivity function was used to compute the eigengene-based connectivity (kME) of each gene. Each module was classified according to region and function using previously known marker genes and GO enrichment analysis.

### 4.7. Network Analysis of Foxg1 and Gata3

The Pearson correlation coefficient between *Foxg1* and *Gata3* and all genes was calculated in the SCT assay count slot of the Seurat object subset of the E13.5 otic epithelial cell subtype (C2). The *p*-value was adjusted by applying Bonferroni correction, and genes with adjusted *p*-values < 0.05 and correlation coefficients > 0.15 were selected. After adding module information obtained from hdWGCNA analysis to the selected genes, the gene network was reconstructed.

### 4.8. Gene Ontology Enrichment Analysis

To confirm the functions of modules and *Foxg1* and *Gata3* network-related genes, gene symbols were mapped to Entrez IDs using org.Mm.eg.db (v.3.19.1) [80]. GO enrichment analysis was conducted using the enrichGO function of ClusterProfiler (v.4.12.0), applying the Benjamini–Hochberg (BH) method (pAdjustMethod = “BH”), where pvalueCutoff = 0.01 and qvalueCutoff = 0.05 criteria, and identifying GO terms for “Biological process (BP)” [81].

### 4.9. Hearing Loss-Associated Gene Enrichment and Correlation Analysis

We used the ClinGen database related to hearing loss, which includes categories such as “Gene”, “Curated Disease Association”, “Inheritance”, and “Classification” [82]. Among the genes included in the database, we found 35 genes that overlapped with those included in each module and sorted diseases by module. The kME values of the genes included in the module were normalized by Z-score. We calculated pairwise correlation coefficients for 35 genes that overlapped between module eigengene and the Clingen database.

### 4.10. Statistical Analysis and Visualization

All analyses and visualizations were performed using the statistical programming language R (v.4.4.1) [83]. Single-cell analysis pipeline and visualization, including Seurat object creation, merging, quality control and filtering, SCTransform normalization, dimensionality reduction (PCA), data integration (harmony), nonlinear dimensionality reduction (UMAP), finding cluster markers, and the assignment of cell types, were performed using built-in functions in Seurat and ggplot2 (v.3.5.1) [77,84]. hdWGCNA pipeline and visualization, including co-expression network analysis, the calculation of module eigengenes and connectivity, as well as the computation of hub gene signature scores employing the UCell method, were performed using built-in functions in hdWGCNA and fmsb (v.0.7.6) [53,85]. Foxg1 and Gata3 network analysis was conducted using Pearson’s correlation and a customized code, followed by Bonferroni correction; then, visualization was performed using Seurat and igraph (v.2.0.3) [77,86]. Gene Ontology enrichment analysis was conducted using built-in functions in ClusterProfiler and Fisher’s exact test, followed by BH correction; enrichplot (v.1.25.0) was employed for visualization. Hearing loss-associated gene enrichment and correlation analysis was conducted using pairwise Pearson’s correlation and a customized code for genes overlapping between module eigengene and the Clingen database; ggplot2 and Complexheatmap (v.2.21.1) were used for visualization [81,87,88]. Schematics of the data analysis pipeline and implementable codes for detailed statistical analysis methods are available at http://github.com/YJkil/Cochlear_Duct (accessed on 26 September 2024).

## Figures and Tables

**Figure 1 ijms-25-12700-f001:**
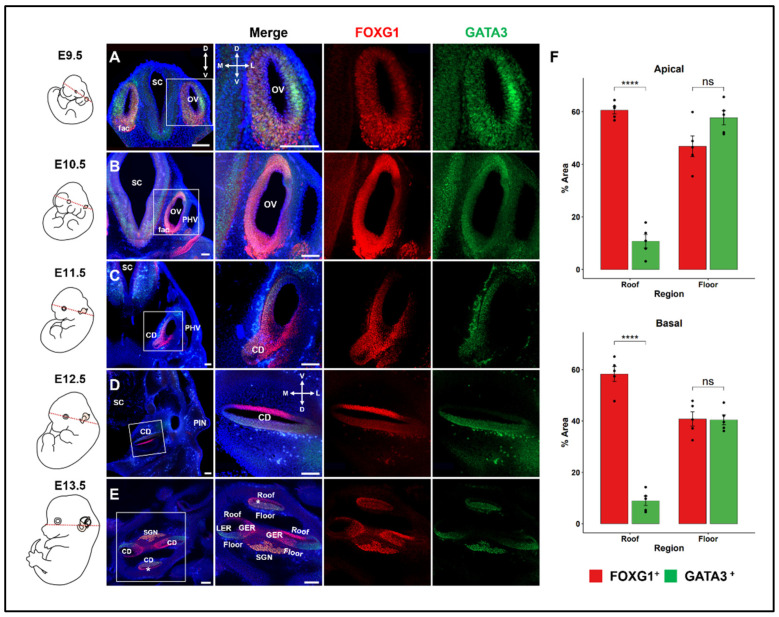
Expressions of FOXG1 and GATA3 in the developing inner ear. At E9.5–E10.5, the otic vesicle begins to extend, and from E11.5, the cochlear duct begins to extend; the cross section of the cochlear duct is observed at E12.5 and E13.5. The section plane is indicated in the embryo image on the left. (**A**,**B**) At E9.5–E10.5, FOXG1 and GATA3 are expressed throughout the otic vesicle. (**C**) In the cochlea at E11.5, *Gata3* shows a change in expression toward the medial–lateral part, whereas *Foxg1* maintains global expression. (**D**) In the cochlea at E12.5, *Gata3* is expressed only in the dorsal region, whereas *Foxg1* continues to be expressed in both the dorsal and ventral regions. (**E**) At E13.5, three cochlear sections and nearby ganglia are visible. In all sections, *Gata3* is expressed at the base, *Foxg1* is expressed throughout the apical duct (asterisk in **E**), and only *Gata3* is expressed at the base. Scale bar = 100 μm. CD, cochlear duct; fac, facio-acoustic (VII-VIII) neural crest complex; GER, the greater epithelial ridge; LER, the lesser epithelial ridge; OV, otic vesicle; PHV, primary head vein; PIN, Pinna; SC, Spinal cord. (**F**) Quantitative analysis of changes in *Foxg1* expression at the base and apex of E13.5 cochlea. The cochlea was divided into the roof and floor, and the proportion of cells expressing *Foxg1* and *Gata3* in the corresponding region was shown (n = 5). At the apex, *Foxg1* expression was high at the roof and floor, but was decreased at the floor of the base compared to the apex. **** *p* ≤ 0.0001.

**Figure 2 ijms-25-12700-f002:**
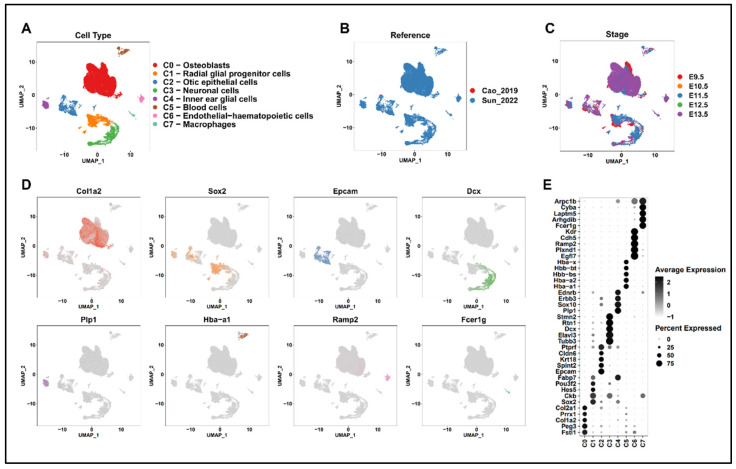
Identification of inner-ear cell type. (**A**–**C**) UMAP plots for cell type (**A**), reference (**B**), and stage (**C**). (**D**) UMAP plots showing marker gene expression in individual clusters. (**E**) Dot plots showing the expressions of the top 5 marker genes and their proportion in individual cell clusters. Color gradient represents the average expression, and dot size indicates the percentage of expressed cells.

**Figure 3 ijms-25-12700-f003:**
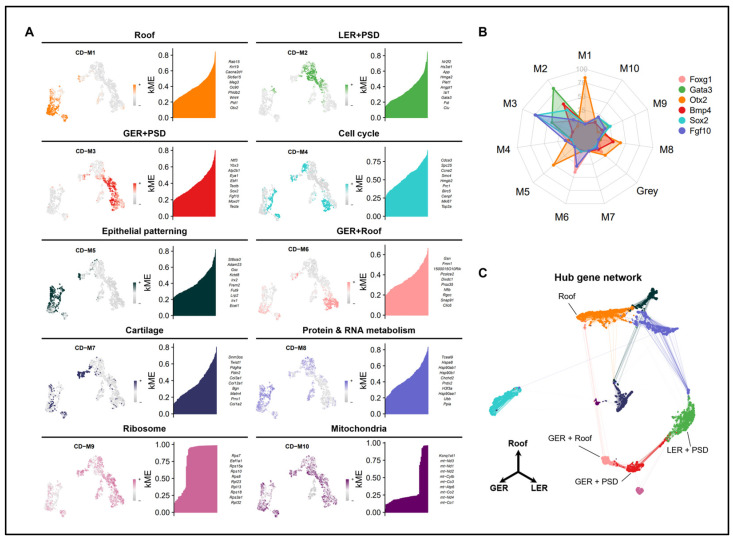
hdWGCNA of cochlear epithelial cell subtype. (**A**) UMAP plots showing modules identified by hdWGCNA and the top 10 hub genes for each module. Colors represent the following modules: CD−M1 (roof) in orange, CD−M2 (LER and PD) in green, CD−M3 (GER and PD) in red, CD−M4 (cell cycle) in cyan, CD−M5 (epithelial patterning) in dark green, CD−M6 (GER and Roof) in light pink, CD−M7 (cartilage) in dark blue, CD−M8 (protein and RNA metabolism) in medium blue, CD−M9 (ribosome) in pink, and CD−M10 (mitochondria) in dark purple. (**B**) Radar plot showing the kME values of hub genes. Colors represent the following genes: *Foxg1* in light pink, *Gata3* in green, *Otx2* in orange, *Bmp4* in red, *Sox2* in cyan, and *Fgf10* in medium blue. (**C**) UMAP plot showing the hub gene network.

**Figure 4 ijms-25-12700-f004:**
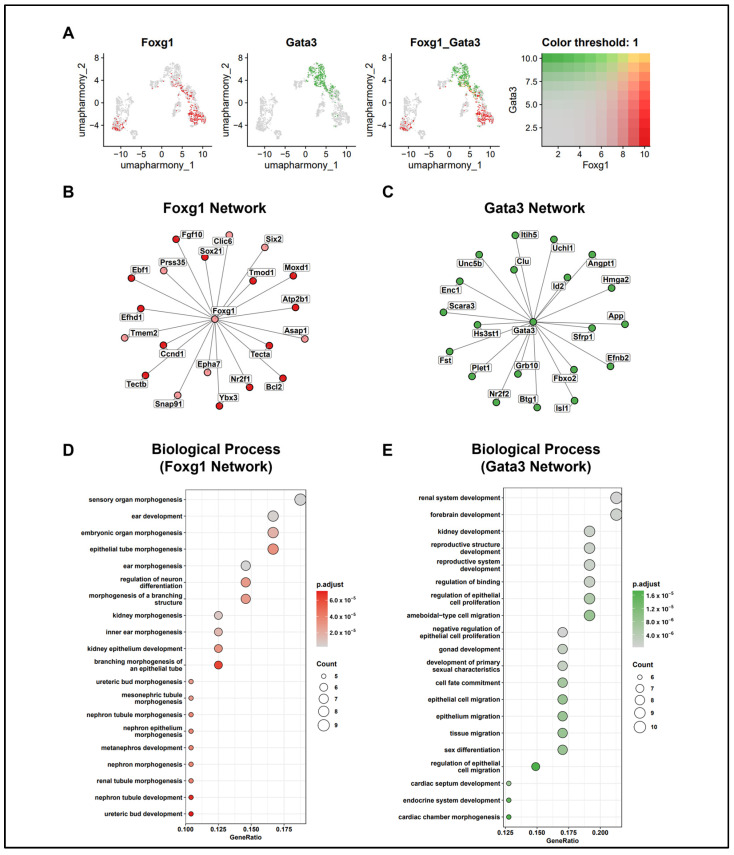
Network analysis of *Foxg1* and *Gata3*. (**A**) UMAP plot showing the expression of *Foxg1* and *Gata3*. Colors represent the following genes: *Foxg1* in red, *Gata3* in green, and cells with blended expression of both genes in yellow. (**B**) Gene network of the top 20 genes highly correlated with *Foxg1*. Colors represent the following modules: CD−M3 (GER and PD) in red and CD−M6 (GER and roof) in light pink. (**C**) Gene network of the top 20 genes highly correlated with *Gata3*. Colors represent the following modules: CD−M2 (LER and PD) in green. (**D**) Dot plot showing the Gene Ontology (GO) terms related to the biological process for genes included in the *Foxg1* network. Dot size represents the gene count, whereas color represents the adjusted *p*-value. (**E**) Dot plot showing the GO terms related to the biological process for genes included in the *Gata3* network. Dot size represents gene count, and color represents the adjusted *p*-value.

**Figure 5 ijms-25-12700-f005:**
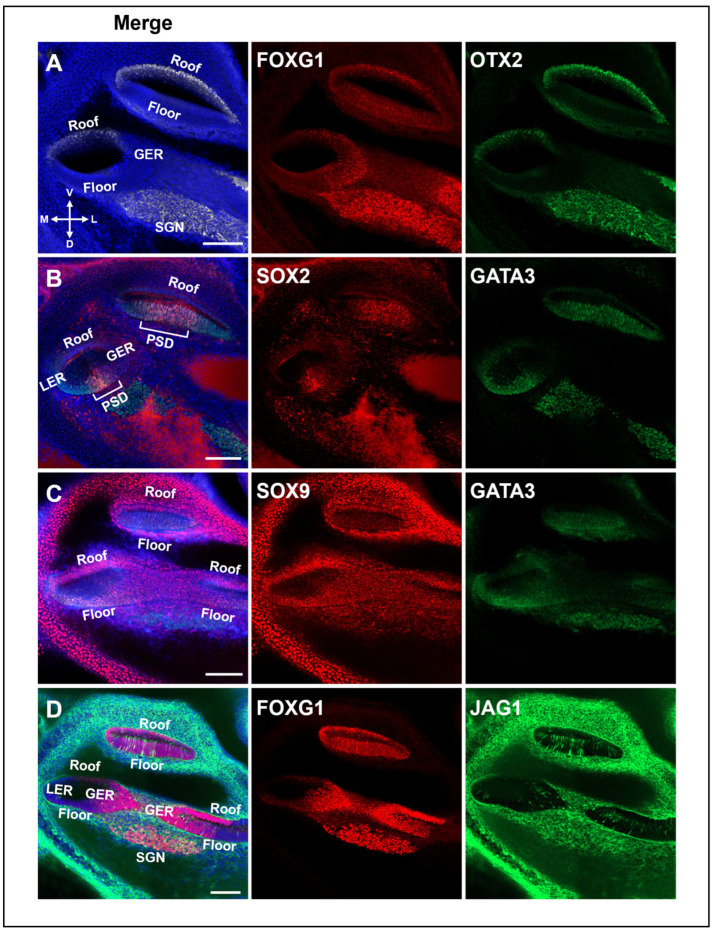
Expression of various marker genes in the E13.5 cochlear duct. (**A**) OTX2, a cochlear roof marker, is specifically expressed in the roof of the apical and primary ducts. FOXG1 is expressed throughout the basal, roof, and GER. In vertex ducts, they are visible on both the roof and the floor. (**B**) SOX2, a sensory domain marker, is co-expressed in the basal region with *Gata3*, and *Gata3* is expressed more broadly throughout the LER. (**C**) Unlike GATA3, SOX9 is expressed throughout the cochlear epithelium. (**D**) The expression of JAG1 can be observed in the prosensory domain and GER, together with that of FOXG1. Scale bar = 100 μm. GER, greater epithelial ridge; LER, lesser epithelial ridge; PSD, prosensory domain; SGN, spiral ganglion neurons.

**Figure 6 ijms-25-12700-f006:**
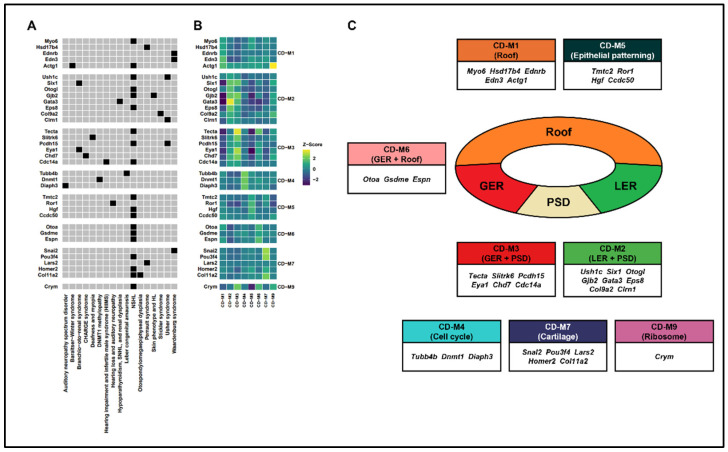
The association between hearing loss-associated disease and modules. (**A**) Heatmap showing genes and hearing loss-related diseases divided by modules. (**B**) Heatmap showing genes and the Z-scored kME values for each module. (**C**) Scheme summarizing the structure of the cochlear duct and genes related to hearing loss diseases within each module.

## Data Availability

The datasets and materials used and/or analyzed during the current study are available from the corresponding author upon reasonable request.

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
