# Peer review of "Molecular Characterization of Subdomain Specification of Cochlear Duct Based on Foxg1 and Gata3"

_ijms, 2024, doi:10.3390/ijms252312700_

Round 1

Reviewer 1 Report

Comments and Suggestions for Authors

This paper is the expression of Foxg1 and Gata3 but requires at least the citation of Duncan’s work (Duncan and Fritzsch, 2013). However, the genetic analysis is great that details many genes that are regulated over time but limit the analysis from E9.5-13.5.  The earliest expression of either Foxg1 or Gata3 deal with the earliest expression of genes necessary for vestibular and cochlear neurons (see Pyott et al., 2024).  With multiple levels requires additional work to fit this paper into the earliest expression of Foxg1 and Gata3.

#32 inner ear or auditory nerve (cranial nerve VIII). Yes, the inner ear consists in the vestibular ganglia and the spiral ganglia (see for example Elliott et al., 2021).  Moreover, the central presentation should have a citations (e.g. Pyott et al., 2024, for example)

#33 We know that the absence of hearing ability is progressing from base to apex (see Elliott et al., Frontiers, 2022) that details the reduction of cochlear HC and spiral ganglia.

#36 update would the work of Shearer and RJH Smith, GeneReviews 2023

#44 a good update would be in the work of Zine and Fritzsch, IJMS, 2023

#49 an update would be in the work of Kopecky et al, Dev Dyn, 2012

#56 an update would be in the Jahan et al., Bioessays, 2015.

#66 an update would the work of Duncan and Fritzsch, 2013

#96-97 Indeed, we already know that Foxg1-cre can eliminate all Gata3 and loses the cochlear hair cells and spiral ganglion neurons (see Duncan and Fritzsch, 2013).  Thanks.

#125 We know that Foxg1 (see Pauley et al., 2006) and Gata3 (see Karis et al., 2001) shows a much earlier of either gene expressions. I suggest to add E8.75

#232 Isl1 is likely interacting with Neurod1 and both are downstream of Neurog1 (see Ma et al., 2000; Filova et al., PNAS, 2022)

#237 It could be too late to get the earliest gene expression for spiral ganglia that are the first to form with vestibular and later cochlear (see Yang et al., Hearing Res, 2011)

#265 Yes, Sox9 interacts with Sox10 (see the work of Szeto et al., PNAS, 2022)

#315 Suggested and update (Pyott et al., 2024).

#322 Yes, Foxg1 causes a shorter cochlear, comparable to nMyc (Kopecky et al., 2021).  However, Foxg1-cre causes a complete loss of all cochlear neurons (Kersigo et al., 2011; Duncan and Fritzsch, 2013) that also depends on Lmx1a/b DKO for cochlear development (Chizhikov et al., 2021) that is somewhat similar in Pax2 deletions (Bouchard et al BMC, 2010). How these genes are needed we want to expand the earliest expression of genes.

#352 suggest presenting a table to show the antibodies first and the secondary second.

#354 one sample is not good enough for E9.5.  In addition, we know that the first neurons form at E8.75 and likely interact with Foxg1 and Gata3.  For that reason, I like to add a E8.75 old mice.

#411 The limit is using in detail for E13.5.  At this stage, many neurons and hair cells are long developed.  For that reason, I like to expand the paper with at least E9.5.

Comments on the Quality of English Language

The language is pretty good with minor problems.

Author Response

This paper is the expression of Foxg1 and Gata3 but requires at least the citation of Duncan’s work (Duncan and Fritzsch, 2013). However, the genetic analysis is great that details many genes that are regulated over time but limit the analysis from E9.5-13.5.  The earliest expression of either Foxg1 or Gata3 deal with the earliest expression of genes necessary for vestibular and cochlear neurons (see Pyott et al., 2024).  With multiple levels requires additional work to fit this paper into the earliest expression of Foxg1 and Gata3.

Comments 1 : [#32 inner ear or auditory nerve (cranial nerve VIII). Yes, the inner ear consists in the vestibular ganglia and the spiral ganglia (see for example Elliott et al., 2021).  Moreover, the central presentation should have a citations (e.g. Pyott et al., 2024, for example)]

Response 1: Thank you for pointing this out. We agree with this comment. Thus, we added the reference as suggested to line #33.

Comments 2 : [#33 We know that the absence of hearing ability is progressing from base to apex (see Elliott et al., Frontiers, 2022) that details the reduction of cochlear HC and spiral ganglia.]

Response 2: Thank you for pointing this out. However, the paper by Elliott et al., Frontiers (2022), deals with hearing loss due to aging, which I think is quite different from the content of this paper.

Comments 3: [#36 update would the work of Shearer and RJH Smith, GeneReviews 2023]

Response 3: Thank you for pointing this out. We agree with this comment. Therefore, we added the reference as suggested to line #37.

Comments 4: [#44 a good update would be in the work of Zine and Fritzsch, IJMS, 2023]

Response 4: Thank you for pointing this out. We agree with this comment. Therefore, we added the reference as suggested to line #44.

Comments 5: [#49 an update would be in the work of Kopecky et al, Dev Dyn, 2012]

Response 5: Thank you for pointing this out. We agree with this comment. Therefore, we added the reference as suggested to line #49.

Comments 6: [#56 an update would be in the Jahan et al., Bioessays, 2015.]

Response 6: Thank you for pointing this out. However, the paper by Jahan et al., Bioessays (2015), largely focuses on hair cell differentiation, which is different from the patterning of the early cochlear duct investigated in this paper. Therefore, I think it is more appropriate to cite the paper by Muthu et al., Development (2019).

Comments 7 : [#66 an update would the work of Duncan and Fritzsch, 2013]

Response 7: Thank you for pointing this out. We agree with this comment. Therefore, we added the reference as suggested to line #66.

Comments 8 : [#96-97 Indeed, we already know that Foxg1-cre can eliminate all Gata3 and loses the cochlear hair cells and spiral ganglion neurons (see Duncan and Fritzsch, 2013).  Thanks.]

Response 8: Thank you for pointing this out. We agree with this comment. Therefore, we added the reference as suggested to line #97.

Comments 9 : [#125 We know that Foxg1 (see Pauley et al., 2006) and Gata3 (see Karis et al., 2001) shows a much earlier of either gene expressions. I suggest to add E8.75]

Response 9: Thank you for pointing this out. However, our study results (Fig. 1) indicated that the mutually exclusive expressions of Foxg1 and gata3 in the base-to-apex axis of the cochlear duct begins at E12.5 and becomes clear at E13.5. Therefore, extending the data to an earlier time period, E8.5, does not seem to support the core findings of this study. In particular, please understand that at E9.5, we confirmed that Foxg1 and Gata3 were co-expressed in most otic vesicles and that the dataset of the paper by Sun et al., Cell Reports (2022), used in this study is provided from E9.5; thus, it is difficult to extend it to E8.5 to integrate our IHC data.

Comments 10: [#232 Isl1 is likely interacting with Neurod1 and both are downstream of Neurog1 (see Ma et al., 2000; Filova et al., PNAS, 2022)]

Response 10: Thank you for pointing this out. We agree with this comment. Therefore, we replaced it with the paper by Filova et al., PNAS (2022), in line 649 ref #58. However, we ask for your understanding that we did not add the paper by Ma et al. (2000) because we judged that it was somewhat different from our results.

Comments 11 : [#237 It could be too late to get the earliest gene expression for spiral ganglia that are the first to form with vestibular and later cochlear (see Yang et al., Hearing Res, 2011)]

Response 11: Thank you for pointing this out. However, as aforementioned, this study elucidated the molecular regulatory network controlling the base-to-apex axis patterning of the cochlear duct, focusing on the transcriptome changes at the time points where Foxg1 and gata3 are mutually exclusive. Although analysis of the entire developmental process of the cochlear duct would be ideal, please understand that the sc-RNA Seq dataset that we could utilize was limited.

Comments 12: [#265 Yes, Sox9 interacts with Sox10 (see the work of Szeto et al., PNAS, 2022)]

Response 12: Thank you for pointing this out. We agree with this comment. Therefore, we replaced it with the paper by Szeto et al., PNAS (2022), in line 660 ref #63.

Comments 13 : [#315 Suggested and update (Pyott et al., 2024).]

Response 13: Thank you for pointing this out. We agree with this comment. Therefore, we added the reference as suggested to line #325.

Comments 14 : [#322 Yes, Foxg1 causes a shorter cochlear, comparable to nMyc (Kopecky et al., 2021).  However, Foxg1-cre causes a complete loss of all cochlear neurons (Kersigo et al., 2011; Duncan and Fritzsch, 2013) that also depends on Lmx1a/b DKO for cochlear development (Chizhikov et al., 2021) that is somewhat similar in Pax2 deletions (Bouchard et al BMC, 2010). How these genes are needed we want to expand the earliest expression of genes.]

Response 14: Thank you for pointing this out. However, as aforementioned, this study elucidated the molecular regulatory network that controls the base-to-apex axis patterning of the cochlear duct. Please understand that we did not cite any other studies that could strengthen the link between the problems in inner ear development caused by the deletion of Foxg1, Lmx1a/b, pax2, etc.

Comments 15: [#352 suggest presenting a table to show the antibodies first and the secondary second.]

Response 15: We sincerely appreciate the reviewer’s comments. However, please understand that we did not present them in a separate table as the types of primary and secondary antibodies utilized in this study were only few.

Comments 16 : [#354 one sample is not good enough for E9.5.  In addition, we know that the first neurons form at E8.75 and likely interact with Foxg1 and Gata3.  For that reason, I like to add a E8.75 old mice.]

Response 16: We sincerely appreciate the reviewer’s comment. However, as the raw data utilized in this study were derived from E9.5-E13.5 embryos, it is difficult to integrate the sc-RNA Seq data with the IHC data for the FoxG1 and Gata3 expressions in E8.75-day embryos. In particular, this paper elucidates the transcriptional regulatory network by the mutually exclusive expressions of FoxG1 and Gata3 and shows that such an exclusive expression is most clearly observed at the E12.5 stage.

Comments 17 : [#411 The limit is using in detail for E13.5.  At this stage, many neurons and hair cells are long developed.  For that reason, I like to expand the paper with at least E9.5.]

Response 17: We sincerely appreciate the reviewer’s comment. However, as this study focuses on elucidating the transcriptome changes caused by the mutually exclusive expressions of Foxg1 and gata3 during the development of the early otic vesicle, we had to limit it to E9.5-E13.5. We ask for the reviewer’s generous understanding.

Comments 18 : [The language is pretty good with minor problems.]

Response 18: The revised version has been re-edited by a professional proofreader and submitted.

Reviewer 2 Report

Comments and Suggestions for Authors

The manuscript depicts the developmental expression pattern of Foxg1 and  Gata3 in mouse cochlear ducts. The study incorporates immunohistochemistry-based imaging of the marker genes and re-analysis of the publicly available scRNA-seq dataset.

This study could be interesting to inner ear developmental biologists.

Comments on the Quality of English Language

The English language is fine. Minor editing in grammar and paraphrasing is needed.

Author Response

Comments 1 : [The manuscript depicts the developmental expression pattern of Foxg1 and  Gata3 in mouse cochlear ducts. The study incorporates immunohistochemistry-based imaging of the marker genes and re-analysis of the publicly available scRNA-seq dataset. This study could be interesting to inner ear developmental biologists. The English language is fine. Minor editing in grammar and paraphrasing is needed.]

Response 1: We sincerely appreciate Reviewer 2’s positive comments on our paper. The revised version has been re-edited by a professional proofreader and submitted.

Reviewer 3 Report

Comments and Suggestions for Authors

This manuscript presents an important developmental study investigating the spatiotemporal expression patterns and roles of Foxg1 and Gata3 in cochlear duct development. The study combines protein expression analysis through immunohistochemistry with single-cell RNA sequencing data analysis, providing novel insights into the molecular mechanisms of cochlear duct development.

Major Strengths:

  1. Clear demonstration of dynamic Foxg1 expression pattern changes from E9.5 to E13.5 through detailed expression analysis.
  2. Successful classification of cochlear development-related gene modules through systematic hdWGCNA analysis.
  3. Establishment of clinical relevance through correlation analysis with hearing loss-related genes.

Specific Revision Suggestions:

  1. For Figures 1-2:
  • Add clear labeling of key structures
  • Include quantitative analysis data for Foxg1 and Gata3 expression intensity
  1. Methods Section:
  • Add more detailed explanation of bioinformatics analysis pipeline
  • Clarify data filtering and quality control criteria
  • Provide detailed statistical analysis methods
  1. Results/Discussion:
  • Expand discussion on functional significance of Foxg1's spatiotemporal expression changes
  • Strengthen discussion of biological significance of identified gene networks
  • Add more specific analysis of correlations with hearing loss-related genes

Implementation of these revisions would significantly enhance the scientific completeness and readability of the manuscript. In particular, the addition of quantitative data and improved clarity in result presentation would be crucial for increasing the reliability of the research findings.

These suggested revisions are achievable and would strengthen the impact and clarity of this valuable developmental biology study. The fundamental findings and conclusions appear sound, but their presentation and interpretation could be enhanced through these modifications.

Author Response

This manuscript presents an important developmental study investigating the spatiotemporal expression patterns and roles of Foxg1 and Gata3 in cochlear duct development. The study combines protein expression analysis through immunohistochemistry with single-cell RNA sequencing data analysis, providing novel insights into the molecular mechanisms of cochlear duct development.

Major Strengths:

  1. Clear demonstration of dynamic Foxg1 expression pattern changes from E9.5 to E13.5 through detailed expression analysis.
  2. Successful classification of cochlear development-related gene modules through systematic hdWGCNA analysis.
  3. Establishment of clinical relevance through correlation analysis with hearing loss-related genes.

Specific Revision Suggestions:

Comments 1 : [For Figures 1-2: Add clear labeling of key structures]

Response 1: Thank you for pointing this out. We agree with this comment. Therefore, we added labels for the main structures in Fig. 1 and added a description of the structure in lines #138–145.

Comments 2: [For Figures 1-2: Include quantitative analysis data for Foxg1 and Gata3 expression intensity]

Response 2: Thank you for pointing this out. We apologize for any confusion caused to the reviewer. The quantification of the expression levels of Foxg1 and Gata3 presented is shown in Fig. 1F, but we confirmed that it was missing from the text and that we have corrected this part in lines #110–112. Furthermore, we added a detailed description of the quantitative analysis to the Materials and Methods section in lines #386–392.

Comments 3: [Methods Section: Add more detailed explanation of bioinformatics analysis pipeline]

Response 3: Thank you for pointing this out. We agree with this comment. Therefore, we added an additional schematic of the bioinformatics analysis pipeline (Supplementary Figure S8), and a detailed description of it is provided in the Materials and Methods section. See line # 404.

Comments 4: [Methods Section: Clarify data filtering and quality control criteria]

Response 4: Thank you for pointing this out. We agree with this comment. Therefore, a detailed description of it is provided in the Materials and Methods section in lines # 408–417.

Comments 5: [Methods Section: Provide detailed statistical analysis methods]

Response 5: Thank you for pointing this out. We agree with this comment. Therefore, a detailed description of it is provided in the Materials and Methods section in lines # 477–495.

Comments 6: [Results/Discussion: Expand discussion on functional significance of Foxg1's spatiotemporal expression changes]

Response 6: Thank you for pointing this out. We agree with this comment. Therefore, we have added an extended discussion of the spatiotemporal expression changes of foxg1 to the Discussion section in lines # 331–341.

Comments 7: [Results/Discussion: Strengthen discussion of biological significance of identified gene networks]

Response 7: Thank you for pointing this out. We agree with this comment. Therefore, we have added an extended discussion of the biological significance of the identified gene networks to the Discussion section in lines # 347–356.

Comments 8: [Results/Discussion: Add more specific analysis of correlations with hearing loss-related genes]

Response 8: Thank you for pointing this out. We agree with this comment. Therefore, a detailed description of it is provided in Supplementary Figure S7 and the Discussion section in lines # 357–367.

Round 2

Reviewer 1 Report

Comments and Suggestions for Authors

That paper has improved.  However, with a few additional suggestions the paper will be more complete.

#98 The work of Dvorakova (2020) showed a much earlier of Foxg1-cre to eliminate Sox2 at E8.5.  Moreover, these data show the absence of cochlear spiral ganglion neurons but has a small population of vestibular that remains.  Given that these recent set of data, I strongly urge to show me the earliest expression at E8.5 mice. 

Fig.1 The position is wrong at E9.5, e10.5 and E11.5.  Flip images to have the same orientation with neurons extrude from ventral, not dorsal as shown here.

# Suggest citing the work of Dvrakova (2020) that uses Fog1-cre to eliminate Sox2.  Of course, no hair cells form and a small population of vestibular form, likely because they are already set up with Neurog1 at E8.5.  Infact, the main impact is the absence of the cochlear duct using a Foxg1-cre; Sox2 cKO.

#359 Ebf1 is interacting with Meis2 (see the work of Koo.....Schimmang, 2024). In fact, the major effect requires using Foxg1-cre to eliminate Meis2.  I strongly cite and expand this presentation.

Author Response

Comments 1 : [#98 The work of Dvorakova (2020) showed a much earlier of Foxg1-cre to eliminate Sox2 at E8.5.  Moreover, these data show the absence of cochlear spiral ganglion neurons but has a small population of vestibular that remains.  Given that these recent set of data, I strongly urge to show me the earliest expression at E8.5 mice.] 

Response 1: Thank you for pointing this out. Thank you for your comment. In addition, I think the paper by Dvorakova (2020) you presented is a very interesting study. However, this study focuses on understanding the patterning process of the cochlear duct by the mutually exclusive expression of Foxg1 and Gata3. In particular, the mutually exclusive expression of Foxg1 and Gata3 began to appear at E12.5 as shown in Fig. 1 and showed a clear pattern at E13.5. Therefore, we utilized the public data set (Sun et al., 2022) that performed sc-RNA-Seq at the E9-E13.5 stage, and we believe that we have sufficiently elucidated the molecular architecture related to the patterning of the cochlear duct that we wanted to know in this study. In addition, please understand that the given 5-day revision period is insufficient to confirm the expression at E8.5.

Comments 2 : [Fig.1 The position is wrong at E9.5, e10.5 and E11.5.  Flip images to have the same orientation with neurons extrude from ventral, not dorsal as shown here.]

Response 2: Thank you for pointing this out. Thank you for your advice. We apologize for the confusion, although we did indicate the direction in the previous Fig1. Therefore, we have revised the images in Fig1A-C.

Comments 3 : [# Suggest citing the work of Dvrakova (2020) that uses Fog1-cre to eliminate Sox2.  Of course, no hair cells form and a small population of vestibular form, likely because they are already set up with Neurog1 at E8.5.  Infact, the main impact is the absence of the cochlear duct using a Foxg1-cre; Sox2 cKO.]

Response 3: Thank you for pointing this out. We agree with this comment. Therefore, we added the reference as suggested to line #265.

Comments 4 : [#359 Ebf1 is interacting with Meis2 (see the work of Koo.....Schimmang, 2024). In fact, the major effect requires using Foxg1-cre to eliminate Meis2.  I strongly cite and expand this presentation.]

Response 4: Thank you for pointing this out. However, our research analysis results did not find any association between ebf1 and meis2. The study by Koo et al (2024) that you suggested did not mention any interaction between ebf1 and meis2. Therefore, it seems unnecessary to cite this paper.

Round 3

Reviewer 1 Report

Comments and Suggestions for Authors

Thanks for clarifying the position in Fig. 1 and citing the work of Dvorakova.